# Riemannian Continuous Normalizing Flows

**Emile Mathieu**[†*], **Maximilian Nickel**[‡]
emile.mathieu@stats.ox.ac.uk, maxn@fb.com
† Department of Statistics, University of Oxford, UK
‡ Facebook Artificial Intelligence Research, New York, USA

## Abstract

Normalizing flows have shown great promise for modelling flexible probability distributions in a computationally tractable way. However, whilst data is often naturally described on Riemannian manifolds such as spheres, tori, and hyperbolic spaces, most normalizing flows implicitly assume a flat geometry, making them either misspecified or ill-suited in these situations. To overcome this problem, we introduce *Riemannian continuous normalizing flows*, a model which admits the parametrization of flexible probability measures on smooth manifolds by defining flows as the solution to ordinary differential equations. We show that this approach can lead to substantial improvements on both synthetic and real-world data when compared to standard flows or previously introduced projected flows.

## 1 Introduction

Learning well-specified probabilistic models is at the heart of many problems in machine learning and statistics. Much focus has therefore been placed on developing methods for modelling and inferring expressive probability distributions. Normalizing flows (Rezende and Mohamed, 2015) have shown great promise for this task as they provide a general and extensible framework for modelling highly complex and multimodal distributions (Papamakarios et al., 2019).

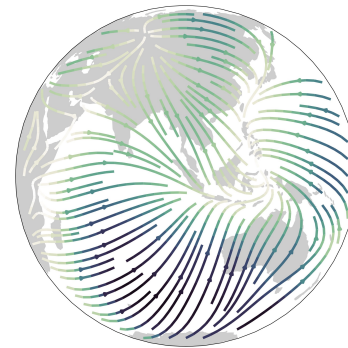

Figure 1: Trajectories generated on the sphere to model volcano eruptions. Note that these converge to the known *Ring of Fire*.

An orthogonal but equally important aspect of well-specified models is to correctly characterize the geometry which describes the proximity of data points. Riemannian manifolds provide a general framework for this purpose and are a natural approach to model tasks in many scientific fields ranging from earth and climate science to biology and computer vision. For instance, storm trajectories may be modelled as paths on the sphere (Karpatne et al., 2019), the shape of proteins can be parametrized using tori (Hamelryck et al., 2006), cell developmental processes can be described through paths in hyperbolic space (Klimovskaia et al., 2020), and human actions can be recognized in video using matrix manifolds (Lui, 2012). If appropriately chosen, manifold-informed methods can lead to improved sample complexity and generalization, improved fit in the low parameter regime, and guide inference methods to interpretable models. They can also be understood as a geometric prior that encodes a practitioner's assumption about the data and imposes an inductive bias.

However, conventional normalizing flows are not readily applicable to such manifold-valued data since their implicit Euclidean assumption makes them unaware of the underlying geometry or borders of the manifold. As a result they would yield distributions having some or all of their mass lying

---

[*]Work done while at Facebook AI research.

outside the manifold, rendering them ill-suited or even misspecified so that central concepts like the reverse Kullback-Leibler (KL) divergence would not even be defined.

In this work, we propose a principled way to combine both of these aspects and parametrize flexible probability distributions on Riemannian manifolds. Specifically, we introduce *Riemmanian continuous normalizing flows* in which flows are defined via vector fields on manifolds and computed as the solution to the associated ordinary differential equation (ODE) (see Figure 1 for an illustration). Intuitively, our method operates by first parametrizing a vector field on the manifold with a neural network, then sampling particles from a base distribution, and finally approximating their flow along the vector field using a numerical solver. Both the neural network and the solver are aware of the underlying geometry which ensures that the flow is always located on the manifold – yielding a *Riemannian* method.

This approach allows us to combine multiple important advantages: One major challenge of normalizing flows lies in designing transformations that enable efficient sampling and density computation. By basing our approach on *continuous normalizing flows (CNFs)* (Chen et al., 2018; Grathwohl et al., 2019; Salman et al., 2018) we avoid strong structural constraints to be imposed on the flow, as is the case for most *discrete* normalizing flows. Such unconstrained *free-form* flows have empirically been shown to be highly expressive (Chen et al., 2019; Grathwohl et al., 2019). Moreover, *projected* methods require a differentiable mapping from a Euclidean space to the manifold, yet such a function cannot be bijective, which in turn leads to numerical challenges. By taking a Riemannian approach, our method is more versatile since it does not rely on an ad-hoc projection map and simultaneously reduces numerical artefacts that interfere with training. To the best of our knowledge, our method is the first to combine these properties as existing methods for normalizing flows on manifolds are either discrete (Bose et al., 2020; Rezende et al., 2020), projected (Gemici et al., 2016; Falorsi et al., 2019; Bose et al., 2020) or manifold-specific (Sei, 2011; Bose et al., 2020; Rezende et al., 2020).

We empirically demonstrate the advantages of our method on constant curvature manifolds – i.e., the Poincaré disk and the sphere – and show the benefits of the proposed approach compared to non-Riemannian and projected methods for maximum likelihood estimation and reverse KL minimization. We also apply our method to density estimation on earth-sciences data (e.g., locations of earthquakes, floods and wildfires) and show that it yields better generalization performance and faster convergence.

## 2 Continuous Normalizing Flows on Riemannian Manifolds

Normalizing flows operate by pushing a simple base distribution through a series of parametrized invertible maps, referred as the *flow*. This can yield a highly complex and multimodal distribution which is typically assumed to live in a Euclidean vector space. Here, we propose a principled approach to extend normalizing flows to manifold-valued data, i.e. *Riemmanian continuous normalizing flows* (RCNFs). Following CNFs (Chen et al., 2018; Grathwohl et al., 2019; Salman et al., 2018) we define manifold flows as the solutions to ODEs. The high-level idea is to parametrize flows through the time-evolution of manifold-valued particles $z$ – in particular via their velocity $\dot{z}(t) = f_\theta(z(t), t)$ where $f_\theta$ denotes a *vector field*. Particles are first sampled from a simple base distribution, and then their evolution is integrated by a manifold-aware numerical solver, yielding a new complex multimodal distribution of the particles. This *Riemannian and continuous* approach has the advantages of allowing almost *free-form* neural networks and of not requiring any mapping from a Euclidean space which would potentially lead to numerical challenges.

For practical purposes, we focus our theoretical and experimental discussion on *constant curvature manifolds* (see Table 1). In addition to being widely used in the literature (Nickel and Kiela, 2017; Davidson et al., 2018; Mardia and Jupp, 2000; Hasnat et al., 2017), these manifolds are convenient to work with since most related geometrical quantities are available in closed-form. However, our

Table 1: Summary of d-dimensional continuous constant (sectional) curvature manifolds.

| Geometry | | Model | Curvature | Coordinates | $\sqrt{\det g} = d\,\mathrm{Vol}\,/d\,\mathrm{Leb}_{\mathbb{R}^d}$ | Compact |
|---|---|---|---|---|---|---|
| **Euclidean** | $\mathbb{R}^d$ | Real vector space | $K = 0$ | Cartesian $z$ | $1$ | No |
| **Hyperbolic** | $\mathbb{B}^d_K$ | Poincaré ball | $K < 0$ | Cartesian $z$ | $\left(2 \,/\, 1 + K\,\|z\|^2\right)^d$ | No |
| **Elliptic** | $\mathbb{S}^d_K$ | Hypersphere | $K > 0$ | n-spherical $\varphi$ | $K^{-\frac{d-1}{2}} \prod_{i=1}^{d-2} \sin(\varphi_i)^{d-i-1}$ | Yes |

proposed approach is generic and could be used on a broad class of manifolds such as product and matrix manifolds like tori and Grassmanians. For a brief overview of relevant concepts in Riemannian geometry please see Appendix A.1 or Lee (2003) for a more thorough introduction.

In the following, we develop the key components which allow us to define continuous normalizing flows that are aware of the underlying Riemannian geometry: flow, likelihood, and vector field.

**Vector flows**   Flows in conventional normalizing flows are defined as smooth mappings $\phi : \mathbb{R}^d \rightarrow \mathbb{R}^d$ which transform a base distribution $z \sim P_0$ into a complex distribution $P_\theta$. For normalizing flows to be well-behaved and convenient to work with, the flow is required to be *bijective* and *differentiable* which introduces significant structural constraints on $\phi$. Continuous normalizing flows overcome this issue by defining the flow $\phi : \mathbb{R}^d \times \mathbb{R} \rightarrow \mathbb{R}^d$ generated by an ordinary differential equation, allowing for unrestricted neural network architectures. Here we show how *vector fields* can be used to define similar flows $\phi : \mathcal{M} \times \mathbb{R} \rightarrow \mathcal{M}$ on general *Riemannian manifolds*.

Consider the temporal evolution of a particle $z(t)$ lying on a d-dimensional manifold $\mathcal{M}$, whose velocity is given by a *vector field* $f_\theta(z(t), t)$. Intuitively, $f_\theta(z(t), t)$ indicates the direction and speed along which the particle is moving on the *manifold's surface*. Classic examples for such vector fields include weathercocks giving wind direction and compasses pointing toward the magnetic north pole of the earth. Formally, let $\mathcal{T}_z\mathcal{M}$ denote the *tangent space* at $z$ and $\mathcal{T}\mathcal{M} = \cap_{z \in \mathcal{M}} \mathcal{T}_z\mathcal{M}$ the associated *tangent bundle*. Furthermore, let $f_\theta : \mathcal{M} \times \mathbb{R} \mapsto \mathcal{T}\mathcal{M}$ denote a vector field on $\mathcal{M}$. The particle's time-evolution according to $f_\theta$ is then given by the following ODE

$$\frac{dz(t)}{dt} = f_\theta(z(t), t). \tag{1}$$

To transform a base distribution using this vector field, we are then interested in a particle's position after time $t$. When starting at an initial position $z(0) = z_0$, the *flow* operator $\phi : \mathcal{M} \times \mathbb{R} \mapsto \mathcal{M}$ gives the particle's position at any time $t$ as $z(t) = \phi(z_0, t)$. Leveraging the fundamental theorem of flows (Lee, 2003), we can show that under mild conditions, this flow is *bijective* and *differentiable*. We write $C^1$ for the set of differentiable functions whose derivative are continuous.

**Proposition 1** (Vector flows). *Let $\mathcal{M}$ be a smooth complete manifold. Furthermore, let $f_\theta$ be a $C^1$-bounded time-dependent vector field. Then there exists a global flow $\phi : \mathcal{M} \times \mathbb{R} \mapsto \mathcal{M}$ such that for each $t \in \mathbb{R}$, the map $\phi(\cdot, t) : \mathcal{M} \mapsto \mathcal{M}$ is a $C^1$-diffeomorphism (i.e. $C^1$ bijection with $C^1$ inverse).*

*Proof.* See Appendix E.1 for a detailed derivation.    □

Note that scaling the vector field as $f_\theta^\alpha \triangleq \alpha f_\theta$ results in a time-scaled flow $\phi^\alpha(z, t) = \phi(z, \alpha t)$. The integration duration $t$ is therefore arbitrary. Without loss of generality we set $t = 1$ and write $\phi \triangleq \phi(\cdot, 1)$. Concerning the evaluation of the flow $\phi$, it generally does no accept a closed-form solution and thus requires to be approximated numerically. To this extent we rely on an explicit and adaptive Runge-Kutta (RK) integrator of order 4 (Dormand and Prince, 1980). However, standard integrators used in CNFs generally do not preserve manifold constraints (Hairer, 2006) . To overcome this issue we rely on a *projective* solver (Hairer, 2011). This solver works by conveniently solving the ODE in the ambient Cartesian coordinates and projecting each step onto the manifold. Projections onto $\mathbb{S}^d$ are computationally cheap since they amount to $l^2$ norm divisions. No projection is required for the Poincaré ball.

**Likelihood**   Having a flow at hand, we are now interested in evaluating the likelihood of our *pushforward* model $P_\theta = \phi_\sharp P_0$. Here, the *pushforward* operator $\sharp$ indicates that one obtains samples $z \sim \phi_\sharp P_0$ as $z = \phi(z_0)$ with $z_0 \sim P_0$. For this purpose, we derive in the following the change in density in terms of the geometry of the manifold and show how to efficiently estimate the likelihood.

*Change in density*   In normalizing flows, we can compute the likelihood of a sample via the change in density from the base distribution to the pushforward. Applying the chain rule we get

$$\log p_\theta(z) - \log p_0(z_0) = \log \left| \det \frac{\partial \phi^{-1}(z)}{\partial z} \right| = -\log \left| \det \frac{\partial \phi(z_0)}{\partial z} \right|. \tag{2}$$

In general, computing the Jacobian's determinant of the flow is challenging since it requires $d$ reverse-mode automatic differentiations to obtain the full Jacobian matrix, and $O(d^3)$ operations to

compute its determinant. CNFs side step direct computation of the determinant by leveraging the time-continuity of the flow and re-expressing Equation 2 as the integral of the *instantaneous change* in log density $\int_0^t \frac{\partial \log p_\theta(z(t))}{\partial t} \, dt$. However, standard CNFs make an implicit Euclidean assumption to compute this quantity which is violated for general Riemannian manifolds. To overcome this issue we express the instantaneous change in log-density in terms of the *Riemannian metric*. In particular, let $G(z)$ denote the *matrix representation of the Riemannian metric* for a given manifold $\mathcal{M}$, then $G(z)$ endows tangent spaces $\mathcal{T}_z\mathcal{M}$ with an inner product. For instance in the Poincaré ball $\mathbb{B}^d$, it holds that $G(z) = (2 \, / \, 1 + K \|z\|^2) \, I_d$, while in Euclidean space $\mathbb{R}^d$ we have $G(z) = I_d$, where $I_d$ denotes the identity matrix. Using the Liouville equation, we can then show that the instantaneous change in variable is defined as follows.

**Proposition 2** (Instantaneous change of variables). *Let $z(t)$ be a continuous manifold-valued random variable given in local coordinates, which is described by the* ODE *from Equation 1 with probability density $p_\theta(z(t))$. The change in log-probability then also follows a differential equation given by*

$$\frac{\partial \log p_\theta(z(t))}{\partial t} = -\operatorname{div}(f_\theta(z(t),t)) = -|G(z(t))|^{-\frac{1}{2}} \operatorname{tr}\left(\frac{\partial \sqrt{|G(z(t))|} f_\theta(z(t),t)}{\partial z}\right) \tag{3}$$

$$= -\operatorname{tr}\left(\frac{\partial f_\theta(z(t),t)}{\partial z}\right) - |G(z(t))|^{-\frac{1}{2}} \left\langle f_\theta(z(t),t), \frac{\partial}{\partial z} \sqrt{|G(z(t))|} \right\rangle. \tag{4}$$

*Proof.* For a detailed derivation of Equation 3 see Appendix C. □

Note that in the Euclidean setting $\sqrt{|G(z)|} = 1$ thus the second term of Equation 4 vanishes and we recover the formula from Grathwohl et al. (2019); Chen et al. (2018).

*Estimating the divergence*    Even though the determinant of Equation 2 has been replaced in Equation 3 by a trace operator with lower computational complexity, we still need to compute the full Jacobian matrix of $f_\theta$. Similarly to Grathwohl et al. (2019); Salman et al. (2018), we make use of Hutchinson's trace estimator to compute the Jacobian efficiently. In particular, Hutchinson (1990) showed that $\operatorname{tr}(A) = \mathbb{E}_{p(\epsilon)}[\epsilon^\top A \epsilon]$ with $p(\epsilon)$ being a $d$-dimensional random vector such that $\mathbb{E}[\epsilon] = 0$ and $\operatorname{Cov}(\epsilon) = I_d$. Leveraging this trace estimator to approximate the divergence in Equation 3 yields

$$\operatorname{div}(f_\theta(z(t),t)) = |G(z(t))|^{-\frac{1}{2}} \mathbb{E}_{p(\epsilon)}\left[\epsilon^\top \frac{\partial \sqrt{|G(z(t))|} f_\theta(z(t),t)}{\partial z} \epsilon\right]. \tag{5}$$

We note that the variance of this estimator can potentially be high since it scales with the inverse of the determinant term $\sqrt{|G(z(t))|}$ (see Appendix D.2). By integrating Equation 3 over time with the stochastic divergence estimator from Equation 5, we get the following total change in log-density between the manifold-valued random variables $z$ and $z_0$

$$\log\left(\frac{p_\theta(z)}{p_0(z_0)}\right) = -\int_0^1 \operatorname{div}(f_\theta(z(t),t)) \, dt = -\mathbb{E}_{p(\epsilon)}\left[\int_0^1 |G(z(t))|^{-\frac{1}{2}} \epsilon^\top \frac{\partial \sqrt{|G(z(t))|} f_\theta(z(t),t)}{\partial z} \epsilon \, dt\right]. \tag{6}$$

It can be seen that Equation 6 accounts again for the underlying geometry through the metric $G(z(t))$. Table 1 lists closed-form solutions of its determinant for constant curvature manifolds. Furthermore, the vector-Jacobian product can be computed through backward auto-differentiation with linear complexity, avoiding the quadratic cost of computing the full Jacobian matrix. Additionally, the integral is approximated via the discretization of the flow returned by the solver.

*Choice of base distribution $P_0$*    The closer the initial base distribution is to the target distribution, the easier the learning task should be. However, it is challenging in practice to incorporate such prior knowledge. We consequently use a uniform distribution on $\mathbb{S}^d$ since it is the most "uncertain" distribution. For the Poincaré ball $\mathbb{B}^d$, we rely on a standard wrapped Gaussian distribution $\mathcal{N}^W$ (Nagano et al., 2019; Mathieu et al., 2019) because it is convenient to work with.

**Vector field**    Finally, we discuss the form of the *vector field* $f_\theta : \mathcal{M} \times \mathbb{R} \to \mathcal{T}\mathcal{M}$ which generates the flow $\phi$ used to pushforward samples. We parametrize $f_\theta$ via a feed-forward neural network which takes as input manifold-valued particles, and outputs their velocities. The architecture of the vector field has a direct impact on the expressiveness of the distribution and is thus crucially important. In order to take into account these geometrical properties we make use of specific input and output layers that we describe below. The rest of the architecture is based on a multilayer perceptron.

*Input layer*   To inform the neural network about the geometry of the manifold $\mathcal{M}$, we use as first layer a *geodesic distance layer* (Ganea et al., 2018; Mathieu et al., 2019) which generalizes *linear layers* to manifolds, and can be seen as computing distances to *decision boundaries* on $\mathcal{M}$. These boundaries are parametrized by *geodesic hyperplanes* $H_{\boldsymbol{w}}$, and the associated *neurons* $h_{\boldsymbol{w}}(\boldsymbol{z}) \propto d_{\mathcal{M}}(\boldsymbol{z}, H_{\boldsymbol{w}})$, with $d_{\mathcal{M}}$ being the geodesic distance. Horizontally stacking several of these *neurons* makes a *geodesic distance layer*. We refer to Appendix E.2 for more details.

*Output layer*   To constrain the neural net to $\mathcal{TM}$, we output vectors in $\mathbb{R}^{d+1}$ when $\mathcal{M} = \mathbb{S}^d$, before projecting them to the tangent space i.e. $\boldsymbol{f}_\theta(\boldsymbol{z}) = \text{proj}_{\mathcal{T}_{\boldsymbol{z}} \mathcal{M}} \, \texttt{neural\_net}(\boldsymbol{z})$. This is not necessary in $\mathbb{B}^d$ since the ambient space is of equal dimension. Yet, velocities scale as $\|\boldsymbol{f}_\theta(\boldsymbol{z})\|_{\boldsymbol{z}} = |G(\boldsymbol{z})|^{1/2} \, \|\boldsymbol{f}_\theta(\boldsymbol{z})\|_2$, hence we scale the $\texttt{neural\_net}$ by $|G(\boldsymbol{z})|^{-1/2}$ s.t. $\|\boldsymbol{f}_\theta(\boldsymbol{z})\|_{\boldsymbol{z}} = \|\texttt{neural\_net}(\boldsymbol{z})\|_2$.

*Regularity*   For the flow to be bijective, the vector field $\boldsymbol{f}_\theta$ is required to be $C^1$ and bounded (cf Proposition 1). The boundness and smoothness conditions can be satisfied by relying on bounded smooth non-linearities in $\boldsymbol{f}_\theta$ such as tanh, along with bounded weight and bias at the last layer.

**Training**   In density estimation and inference tasks, one aims to learn a model $P_\theta$ with parameters $\theta$ by minimising a *divergence* $\mathcal{L}(\theta) = D(P_{\mathcal{D}} \parallel P_\theta)$ w.r.t. a target distribution $P_{\mathcal{D}}$. In our case, the parameters $\theta$ refer to the parameters of the vector field $\boldsymbol{f}_\theta$. We minimize the loss $\mathcal{L}(\theta)$ using first-order stochastic optimization, which requires Monte Carlo estimates of loss gradients $\nabla_\theta \mathcal{L}(\theta)$. We back-propagate gradients through the explicit solver with $O(1/h)$ memory cost, $h$ being the step size. When the loss $\mathcal{L}(\theta)$ is expressed as an expectation over the model $P_\theta$, as in the *reverse* KL divergence, we rely on the *reparametrization trick* (Kingma and Welling, 2014; Rezende et al., 2014). In our experiments we will consider both the negative log-likelihood and reverse KL objectives

$$\mathcal{L}^{\text{Like}}(\theta) = -\mathbb{E}_{\boldsymbol{z} \sim P_{\mathcal{D}}} [\log p_\theta(\boldsymbol{z})] \text{ and } \mathcal{L}^{\text{KL}}(\theta) = D_{\text{KL}} (P_\theta \parallel P_{\mathcal{D}}) = \mathbb{E}_{\boldsymbol{z} \sim P_\theta} [\log p_\theta(\boldsymbol{z}) - \log p_{\mathcal{D}}(\boldsymbol{z})]. \quad (7)$$

Additionally, regularization terms can be added in the hope of improving training and generalization. See Appendix D for a discussion and connections to the dynamical formulation of optimal transport.

## 3   Related work

Here we discuss previous work that introduced normalizing flows on manifolds. For clarity we split these into *projected* vs *Riemannian* methods which we describe below.

**Projected methods**   These methods consist in parametrizing a normalizing flow on $\mathbb{R}^d$ and then pushing-forward the resulting distribution along an invertible map $\psi : \mathbb{R}^d \to \mathcal{M}$. Yet, the existence of such an *invertible* map is equivalent to $\mathcal{M}$ being homeomorphic to $\mathbb{R}^d$ (e.g. being "flat"), hence limiting the scope of that approach. Moreover there is no principled way to choose such a map, and different choices lead to different numerical or computational challenges which we discuss below.

*Exponential map*   The first generic *projected* map that comes to mind in this setting is the exponential map $\exp_{\boldsymbol{\mu}} : T_{\boldsymbol{\mu}} \mathcal{M} \cong \mathbb{R}^d \to \mathcal{M}$, which parameterizes geodesics starting from $\boldsymbol{\mu}$ with velocity $\boldsymbol{v} \in T_{\boldsymbol{\mu}} \mathcal{M}$. This leads to so called *wrapped* distributions $P_\theta^{\text{W}} = \exp_{\boldsymbol{\mu}\sharp} P$, with $P$ a probability measure on $\mathbb{R}^d$. This approach has been taken by Falorsi et al. (2019) to parametrize probability distributions on Lie groups. Yet, in compact manifolds – such as spheres or the SO(3) group – computing the density of *wrapped* distributions requires an infinite summation, which in practice needs to be truncated. This is not the case however on hyperbolic spaces (like the Poincaré ball) since the exponential map is bijective on these manifolds. This approach has been proposed in Bose et al. (2020) where they extend Real-NVP (Dinh et al., 2017) to the hyperboloid model of hyperbolic geometry. In addition to this *wrapped* Real-NVP, they also introduced a hybrid coupling model which is empirically shown to be more expressive. We note however that the exponential map is believed to be "badly behaved" away from the origin (Dooley and Wildberger, 1993; Al-Mohy and Higham, 2010).

*Stereographic map*   Alternatively to the exponential map, Gemici et al. (2016) proposed to parametrize probability distributions on $\mathbb{S}^d$ via the *stereographic projection* defined as $\rho(\boldsymbol{z}) = \boldsymbol{z}_{2:d} \, / \, (1 + z_1)$ with *projection point* $-\{\boldsymbol{\mu}_0\} = (-1, 0, \ldots, 0)$. Gemici et al. then push a probability measure $P$ defined on $\mathbb{R}^d$ along the inverse of the *stereographic* map $\rho$, yielding $P_\theta^{\text{S}} = \rho_\sharp^{-1} P$.

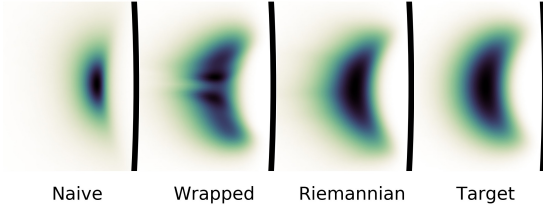

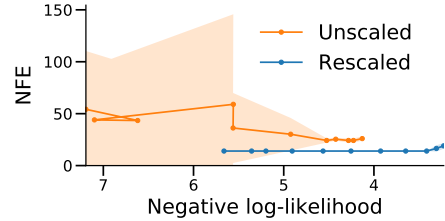

Naive  Wrapped  Riemannian  Target

Figure 2: Probability densities on $\mathbb{B}^2$. Models have been trained by maximum likelihood to fit $\mathcal{N}^{\mathrm{W}}(\exp_{\mathbf{0}}(2\,\partial x), \Sigma)$. The black semi-circle indicates the disk's border. The best run out of twelve trainings is shown for each model.

Figure 3: Ablation study of the vector field architecture for the *Riemannian* model. Models have been trained to fit a $\mathcal{N}^{\mathrm{W}}(\exp_{\mathbf{0}}(\partial x), \Sigma)$.

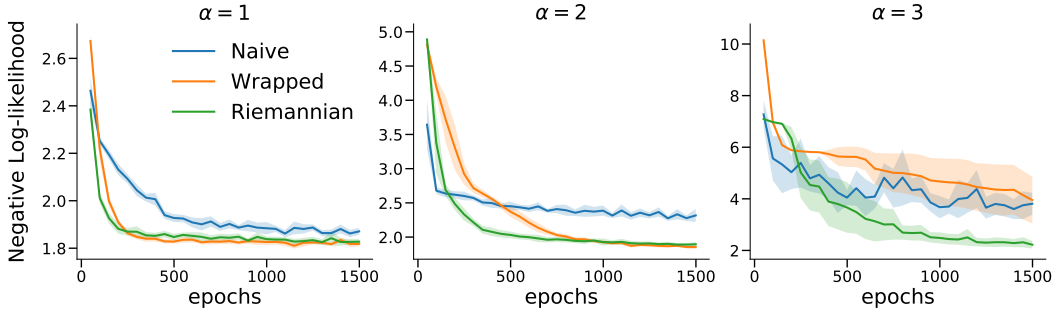

Figure 4: Negative Log-likelihood of CNFs trained to fit a $\mathcal{N}^{\mathrm{W}}(\exp_{\mathbf{0}}(\alpha\,\partial x), \Sigma)$ target on $\mathbb{B}^2$.

However, the stereographic map $\rho$ is not injective, and projects $-\boldsymbol{\mu}_0$ to $\infty$. This implies that spherical points close to the projection point $-\{\boldsymbol{\mu}_0\}$ are mapped far away from the origin of the plane. Modelling probability distributions with mass close to $\{-\boldsymbol{\mu}_0\}$ may consequently be numerically challenging since the norm of the Euclidean flow would explode. Similarly, Rezende et al. (2020) introduced flows on hyperspheres and tori by using the inverse tangent function. Although this method is empirically shown to perform well, it similarly suffers from numerical instabilities near singularity points.

**Riemannian methods**  In contrast to *projected* methods which rely on mapping the manifold to a Euclidean space, *Riemannian* methods do not. As a consequence they side-step any artefact or numerical instability arising from the manifold's projection. Early work (Sei, 2011) proposed transformations along geodesics on the hypersphere by evaluating the exponential map at the gradient of a scalar manifold function. Recently, Rezende et al. (2020) introduced ad-hoc *discrete Riemannian* flows for hyperspheres and tori based on Möbius transformations and spherical splines. We contribute to this line of work by introducing *continuous* flows on general Riemannian manifolds. In contrast to *discrete* flows (e.g. Bose et al., 2020; Rezende et al., 2020), *time-continuous* flows as ours alleviate strong structural constraints on the flow by implicitly parametrizing it as the solution to an ODE (Grathwohl et al., 2019). Additionally, recent and concurrent work (Lou et al., 2020; Falorsi and Forré, 2020) proposed to extend neural ODEs to smooth manifolds.

## 4   Experimental results

We evaluate the empirical performance of the above-mentioned models on hyperbolic and spherical geometry. We will first discuss experiments on two synthetic datasets where we highlight specific pathologies of the naive and projected methods via unimodal distributions at the point (or the limit) of the pathology. This removes additional modelling artefacts that would be introduced through more complex distributions and allows to demonstrate advantages of our approach on the respective manifolds. We further show that these advantages also translate to substantial gains on highly multi-modal real world datasets.

For all *projected* models (e.g. stereographic and wrapped cf Section 3), the vector field's architecture is chosen to be a multilayer perceptron as in Grathwohl et al. (2019), whilst the architecture described in Section 2 is used for our *Riemannian* (continuous normalizing flow) model. For fair comparisons,

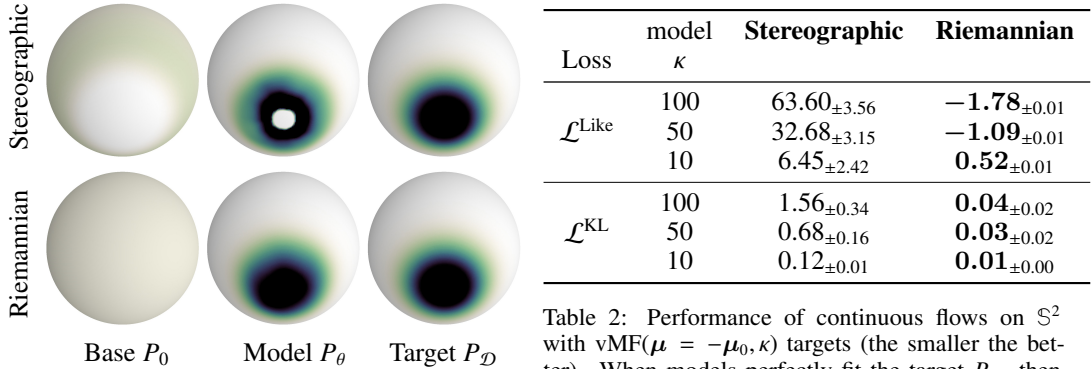

Figure 5: Probability distributions on $\mathbb{S}^2$. Models trained to fit a vMF($\mu = -\mu_0, \kappa = 10$).

Table 2: Performance of continuous flows on $\mathbb{S}^2$ with vMF($\mu = -\mu_0, \kappa$) targets (the smaller the better). When models perfectly fit the target $P_{\mathcal{D}}$, then $\mathcal{L}^{\text{Like}} = \mathbb{H}[P_{\mathcal{D}}]$ which decreases with $\kappa$, explaining $\mathcal{L}^{\text{Like}}$'s results for the *Riemannian* model.

| Loss | model $\kappa$ | **Stereographic** | **Riemannian** |
|---|---|---|---|
| $\mathcal{L}^{\text{Like}}$ | 100 | $63.60_{\pm 3.56}$ | $\mathbf{-1.78}_{\pm 0.01}$ |
| | 50 | $32.68_{\pm 3.15}$ | $\mathbf{-1.09}_{\pm 0.01}$ |
| | 10 | $6.45_{\pm 2.42}$ | $\mathbf{0.52}_{\pm 0.01}$ |
| $\mathcal{L}^{\text{KL}}$ | 100 | $1.56_{\pm 0.34}$ | $\mathbf{0.04}_{\pm 0.02}$ |
| | 50 | $0.68_{\pm 0.16}$ | $\mathbf{0.03}_{\pm 0.02}$ |
| | 10 | $0.12_{\pm 0.01}$ | $\mathbf{0.01}_{\pm 0.00}$ |

we also parametrize *projected* models with a CNF. Also, all models are chosen to have approximately the same number of parameters. All models were implemented in PyTorch (Paszke et al., 2017) and trained by stochastic optimization with Adam (Kingma and Ba, 2015). All 95% confidence intervals are computed over 12 runs. Please refer to Appendix G for full experimental details.

**Hyperbolic geometry and limits of conventional and wrapped methods** First, we aim to show that conventional normalizing flows are ill-suited for modelling target manifold distributions. These are blind to the geometry, so we expect them to behave poorly when the target is located where the manifold behaves most differently from a Euclidean space. We refer to such models as *naive* and discuss their properties in more detail in Appendix B.1. Second, we wish to inspect the behaviour of *wrapped* models (see Section 3) when the target is away from the exponential map origin.

To this extent we parametrize a wrapped Gaussian target distribution $\mathcal{N}^{\text{W}}(\exp_0(\alpha \; \partial x), \Sigma) = \exp_{\mu\sharp} \mathcal{N}(\alpha \; \partial x, \Sigma)$ defined on the Poincaré disk $\mathbb{B}^2$ (Nagano et al., 2019; Mathieu et al., 2019). The scalar parameter $\alpha$ allows us to locate the target closer or further away from the origin of the disk. We put three CNFs models on the benchmark; our *Riemannian* (from Section 2), a conventional *naive* and a *wrapped* model. The base distribution $P_0$ is a standard Gaussian for the *naive* and *wrapped* models, and a standard wrapped Gaussian for the *Riemannian* model. Models are trained by maximum likelihood until convergence. Throughout training, the Hutchinson's estimator is used to approximate the divergence as in Equation 5. It can be seen from Figure 4 that the *Riemannian* model indeed outperforms the *naive* and *wrapped* models as we increase the values of $\alpha$ – i.e., the closer we move to the boundary of the disk. Figure 2 shows that qualitatively the *naive* and *wrapped* models seem to indeed fail to properly fit the target when it is located far from the origin. Additionally, we assess the architectural choice of the vector field used in our *Riemannian* model. In particular, we conduct an ablation study on the rescaling of the output layer, by training for 10 iterations a *rescaled* and an *unscaled* version of our model. Figure 3 shows that the number of function evaluations (NFE) tends to be large and sometimes even dramatically diverges when the vector field's output is *unscaled*. In addition to increasing the computational cost, this in turns appears to worsen the convergence's speed of the model. This further illustrates the benefits of our vector field parameterization.

**Spherical geometry and limits of the stereographic projection model** Next, we evaluate the ability of our model and the stereographic projection model from Section 3 to approximate distributions on the sphere which are located around the projection point $-\mu_0$. We empirically assess this phenomenon by choosing the target distribution to be a Von-Mises Fisher (Downs, 1972) distribution vMF($\mu, \kappa$) located at $\mu = -\mu_0$, and with concentration $\kappa$ (which decreases with the variance). Along with the *stereographic* projection method, we also consider our *Riemannian* model from Section 2. We neither included the *naive* model since it is misspecified here (leading to an undefined reverse KL divergence), nor the *wrapped* model as computing its density requires an infinite summation (see Section 3). The base distribution $P_0$ is chosen to be a standard Gaussian on $\mathbb{R}^2$ for the *stereographic* model and a uniform distribution on $\mathbb{S}^2$ for the *Riemannian* model. Models are trained by computing the exact divergence. The performance of these two models are quantitatively assessed on both the negative log-likelihood and reverse KL criteria.

Table 3: Negative test log-likelihood of continuous normalizing flows on $\mathbb{S}^2$ datasets.

| | Volcano | Earthquake | Flood | Fire |
|---|---|---|---|---|
| **Mixture vMF** ■ | $-0.31_{\pm 0.07}$ | $0.59_{\pm 0.01}$ | $1.09_{\pm 0.01}$ | $-0.23_{\pm 0.02}$ |
| **Stereographic** ■ | $-0.64_{\pm 0.20}$ | $0.43_{\pm 0.04}$ | $0.99_{\pm 0.04}$ | $-0.40_{\pm 0.06}$ |
| **Riemannian** ■ | $-0.97_{\pm 0.15}$ | $\mathbf{0.19_{\pm 0.04}}$ | $\mathbf{0.90_{\pm 0.03}}$ | $\mathbf{-0.66_{\pm 0.05}}$ |
| Learning curves | 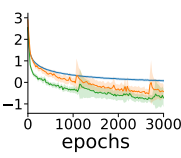 |  |  |  |
| Data size | 829 | 6124 | 4877 | 12810 |

Figure 5 shows densities of the target distribution along with the base and learned distributions. We observe that the stereographic model fails to push mass close enough to the singularity point $-\mu_0$, as opposed to the Riemannian model which perfectly fits the target. Table 2 shows the negative log-likelihood and reverse KL losses of both models when varying the concentration parameter $\kappa$ of the vMF target. The larger the concentration $\kappa$ is, the closer to the singularity point $-\mu_0$ the target's mass gets. We observe that the *Riemannian* model outperforms the *stereographic* one to fit the target for both objectives, although this performance gap shrinks as the concentration gets smaller. Also, we believe that the gap in performance is particularly large for the log-likelihood objective because it heavily penalizes models that fail to cover the support of the target. When the vMF target is located away from the singularity point, we noted that both models were performing similarly well.

**Density estimation of spherical data**   Finally, we aim to measure the expressiveness and modelling capabilities of our method on real world datasets. To this extent, we gathered four earth location datasets, representing respectively volcano eruptions (NOAA, 2020b), earthquakes (NOAA, 2020a), floods (Brakenridge, 2017) and wild fires (EOSDIS, 2020). We approximate the earth's surface (and thus also these data points) as a perfect sphere. Along our *Riemannian* CNF, we also assess the fitting capacity of a mixture of von Mises-Fisher (vMF) distributions and a *stereographic* projected CNF. The locations of the vMF components are learned via stochastic Riemannian optimization (Bonnabel, 2013; Becigneul and Ganea, 2019). The learning rate and number of components are selected by hyperparameter grid search. In our experiments, we split datasets randomly into training and testing datasets, and fit the models by maximum likelihood estimation on the training dataset. CNF models are trained by computing the exact divergence.

We observe from Table 3 that for all datasets, the *Riemannian* model outperforms its *stereographic* counterpart and the mixture of vMF distributions by a large margin. It can also be seen from the learning curves that the *Riemannian* model converges faster. Figure 6 shows the learned spherical distributions along with the training and testing datasets. We note that qualitatively the *stereographic* distribution is generally more diffuse than its *Riemannian* counterpart. It also appears to allocate some of its mass outside the target support, and to cover less of the data points. Additional figures are shown in Appendix H.

**Limitations**   In the spherical setting, the stochastic estimator to approximate the divergence from Equation 5 exhibits high variance. Its variance scales with the inverse of $\sqrt{|G(z)|} = \sin(\theta)$, which becomes too large around the north pole and thus requires the use of the exact estimator. On large-scale datasets, where the stochastic estimator has important runtime advantages, this issue could be alleviated by choosing a different vector field basis than the one induced by some local coordinates (e.g. Falorsi and Forré, 2020). For the Poincaré ball, no such variance behavior of the stochastic estimator exists and it can readily be applied to large-scale data.

Compared to a well-optimized linear layer, the use of the geodesic distance layer (see Section 2) induces an extra computational cost as shown in Figure 7. Empirically, the geodesic layer helps to improve performance in the hyperbolic setting but had less of an effect in the spherical setting. As

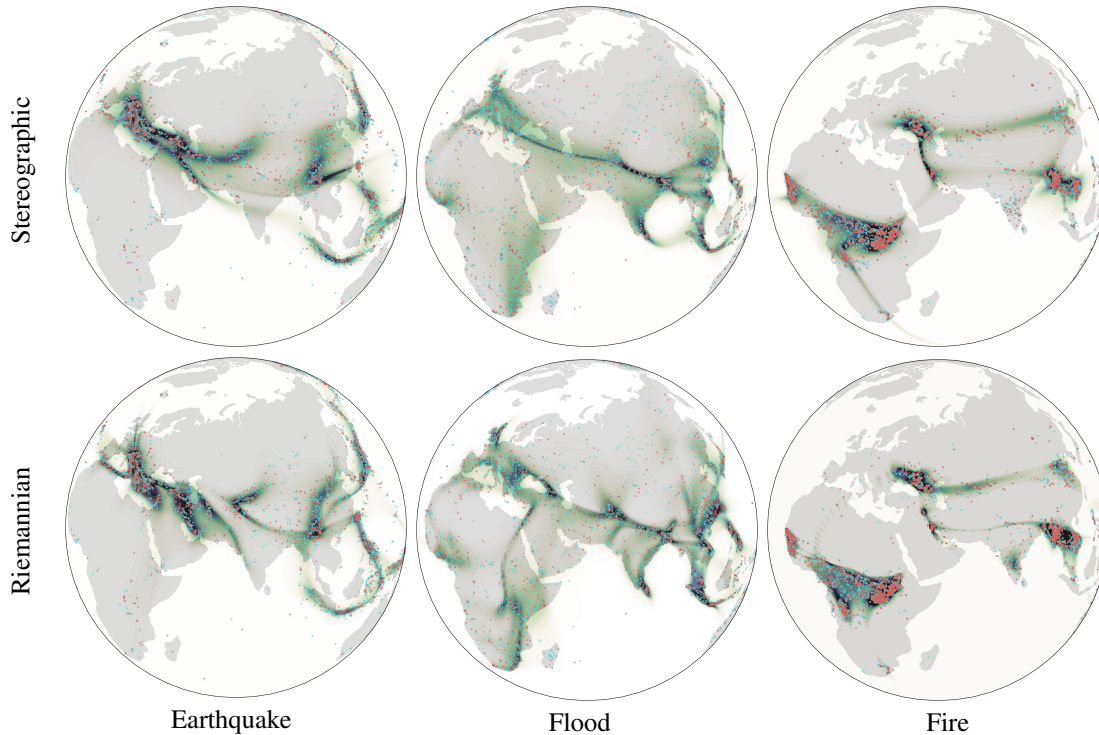

Figure 6: Density estimation for earth sciences data. Blue and red dots represent training and testing datapoints, respectively. Heatmaps depict the log-likelihood of the trained models.

such, the geodesic layer can be regarded as an optional component that can improve the quality of the model at an additional computational cost.

# 5 Discussion

In this paper we proposed a principled way to parametrize expressive probability distributions on Riemannian manifolds. Specifically, we introduced *Riemmanian continuous normalizing flows* in which flows are defined via vector fields on manifolds and computed as the solution to the associated ODE. We empirically demonstrated that this method can yield substantial improvements when modelling data on constant curvature manifolds compared to conventional or projected flows.

## Broader impact

The work presented in this paper focuses on the learning of well-specified probabilistic models for manifold-valued data. Consequently, its applications are especially promising to advance scientific understanding in fields such as earth and climate science, computational biology, and computer vision. As a foundational method, our work inherits the broader ethical aspects and future societal consequences of machine learning in general.

## Acknowledgments

We are grateful to Adam Foster, Yann Dubois, Laura Ruis, Anthony Caterini, Adam Golinski, Chris Maddison, Salem Said, Alessandro Barp, Tom Rainforth and Yee Whye Teh for fruitful discussions and support. EM research leading to these results received funding from the European Research Council under the European Union's Seventh Framework Programme (FP7/2007- 2013) ERC grant agreement no. 617071 and he acknowledges Microsoft Research and EPSRC for funding EM's studentship.

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
