[Supplementary Material]

# Appendix for

# Riemannian Continuous Normalizing Flows

## A   Constant curvature manifolds

In the following, we provide a brief overview of Riemannian geometry and constant curvature manifolds, specifically the Poincaré ball and the hypersphere models. We will use $\|\cdot\|$ and $\langle \cdot, \cdot \rangle$ to denote the Euclidean norm and inner product. For norms and inner products on tangent spaces $\mathcal{T}_z \mathcal{M}$, we write $\|\cdot\|_z$ and $\langle \cdot, \cdot \rangle_z$ where $z \in \mathcal{M}$.

### A.1   Review of Riemannian geometry

A real, smooth *manifold* $\mathcal{M}$ is a set of points $z$, which is "locally similar" to a linear space. For every point $z$ of the manifold $\mathcal{M}$ is attached a real vector space of the same dimensionality as $\mathcal{M}$ called the *tangent space* $\mathcal{T}_z \mathcal{M}$. Intuitively, it contains all the possible directions in which one can tangentially pass through $z$. Taking the disjoint union of all tangent spaces yields the *tangent bundle* $\mathcal{T} \mathcal{M} = \cap_{z \in \mathcal{M}} \mathcal{T}_z \mathcal{M}$. For each point $z$ of the manifold, the *metric tensor* $\mathfrak{g}(z)$ defines an inner product on the associated tangent space as $\mathfrak{g}(z) = \langle \cdot, \cdot \rangle_z : \mathcal{T}_z \mathcal{M} \times \mathcal{T}_z \mathcal{M} \to \mathbb{R}$. The *matrix representation of the Riemannian metric $G(z)$*, is defined such that

$$\forall u, v \in \mathcal{T}_z \mathcal{M} \times \mathcal{T}_z \mathcal{M}, \ \langle u, v \rangle_z = \mathfrak{g}(z)(u, v) = u^T G(z) v.$$

A *Riemannian manifold* is then given as a tuple $(\mathcal{M}, \mathfrak{g})$ (Petersen, 2006). The metric tensor gives a *local* notion of angle, length of curves, surface area and volume, from which *global* quantities can be derived by integrating local contributions. A norm is induced by the inner product on $\mathcal{T}_z \mathcal{M}$: $\|\cdot\|_z = \sqrt{\langle \cdot, \cdot \rangle_z}$. An infinitesimal volume element is induced on each tangent space $\mathcal{T}_z \mathcal{M}$, and thus a measure $d\mathrm{Vol}(z) = \sqrt{|G(z)|} \, d\mathrm{Leb}(z)$ on the manifold, with $\mathrm{Leb}(z)$ being the Lebesgue measure. The length of a curve $\gamma : t \mapsto \gamma(t) \in \mathcal{M}$ is given by $L(\gamma) = \int_0^1 \|\gamma'(t)\|_{\gamma(t)} dt$. The concept of straight lines can then be generalized to *geodesics*, which are constant speed curves giving the shortest path between pairs of points $z, y$ of the manifold: $\gamma^* = \arg\min L(\gamma)$ with $\gamma(0) = z$, $\gamma(1) = y$ and $\|\gamma'(t)\|_{\gamma(t)} = 1$. A *global* distance is thus induced on $\mathcal{M}$ given by

$$d_{\mathcal{M}}(z, y) = \inf L(\gamma).$$

Endowing $\mathcal{M}$ with that distance consequently defines a metric space $(\mathcal{M}, d_{\mathcal{M}})$. The concept of moving along a "straight" curve with constant velocity is given by the *exponential map*. In particular, there is a unique unit speed *geodesic* $\gamma$ satisfying $\gamma(0) = z$ with initial tangent vector $\gamma'(0) = v$. The corresponding exponential map is then defined by $\exp_z(v) = \gamma(1)$. The *logarithm map* is the inverse $\log_z = \exp_z^{-1} : \mathcal{M} \to \mathcal{T}_z \mathcal{M}$. The $\exp_z$ map is well-defined on the full tangent space $\mathcal{T}_z \mathcal{M}$ for all $z \in \mathcal{M}$ if and only if $\mathcal{M}$ is geodesically complete, i.e. if all geodesics can "run" indefinitely. This is the case for the Poincaré ball and hypersphere.

### A.2   The Poincaré ball model of hyperbolic geometry

In the following, we provide a brief overview of key concepts related to hyperbolic geometry. A $d$-dimensional *hyperbolic* space is a complete, simply connected, $d$-dimensional Riemannian manifold with *constant negative curvature $K$*. The *Poincaré ball* is one model of this geometry, and is formally defined as the Riemannian manifold $\mathbb{B}_K^d = (\mathcal{B}_K^d, \mathfrak{g}_K)$. Here $\mathcal{B}_K^d$ denotes the open ball of radius $1/\sqrt{|K|}$, and $\mathfrak{g}_K$ the *metric tensor* $\mathfrak{g}_K(z) = (\lambda_z^K)^2 \mathfrak{g}^e(z)$, where $\lambda_z^K = \frac{2}{1 + K\|z\|^2}$ and $\mathfrak{g}^e$ denotes the Euclidean metric tensor, i.e. the usual dot product. The induced *invariant measure* Vol is absolutely continuous with respect to the Lebesgue measure Leb, and its density is given by $\frac{d\mathrm{Vol}}{d\mathrm{Leb}}(z) = \sqrt{|G(z)|} = (\lambda_z^K)^d$ for all $z \in \mathbb{B}_K^d$. As motivated by Skopek et al. (2020), the Poincaré ball $\mathbb{B}_K^d$ can conveniently be described through the formalism of *gyrovector spaces* (Ungar, 2008). These can be seen as an analogy to the way vector spaces are used in Euclidean geometry, but in the non-Euclidean geometry setting. In

particular, the *Möbius addition* $\oplus_K$ of $z, y$ in $\mathbb{B}_K^d$ is defined as

$$z \oplus_K y = \frac{(1 - 2K \langle z, y \rangle - K\|y\|^2)z + (1 + K\|z\|^2)y}{1 - 2K \langle z, y \rangle + K^2\|z\|^2\|y\|^2}.$$

Then the *exponential map* can be expressed via this *Möbius addition* as

$$\exp_z^K(v) = z \oplus_K \left( \tanh \left( \sqrt{-K} \frac{\lambda_z^K \|v\|}{2} \right) \frac{v}{\sqrt{-K}\|v\|} \right)$$

where $x = -z \oplus_K y$ for all $x, y \in \mathbb{B}_K^d$.

### A.3 The hypersphere model of elliptic geometry

In the following, we discuss key concepts related to positively curved spaces known as *elliptic* spaces, and in particular to the *hypersphere* model. The d-sphere, or *hyperphere*, is a compact submanifold of $\mathbb{R}^{d+1}$ with positive constant curvature $K$ whose support is defined by $\mathcal{S}_K^d = \{z \in \mathbb{R}^{d+1} \mid \langle z, z \rangle = 1/K\}$. It is endowed with the pull-back metric of the ambient Euclidean space.

**Sphere**   In the two-dimensional setting $d = 2$, we rely on polar coordinates to parametrize the sphere $\mathbb{S}^2$. These coordinates consist of polar $\theta \in [0, \pi]$ and azimuth $\varphi \in [0, 2\pi)$ angles. The ambient Cartesian coordinates are then given by $r(\theta, \varphi) = (\sin(\theta)\cos(\varphi), \sin(\theta)\sin(\varphi), \cos(\theta))$. We have $\sqrt{|G(\theta, \varphi)|} = \sin(\theta)$. Applying the generic divergence formula (see Equation 11) yields the celebrated spherical divergence formula

$$\operatorname{div}(g) = \frac{1}{\sin(\theta)} \frac{\partial}{\partial \theta} \left( \sin(\theta) \, g^\theta(\theta, \varphi) \right) + \frac{1}{\sin(\theta)} \frac{\partial}{\partial \varphi} \left( g^\varphi(\theta, \varphi) \right).$$

**Hypersphere**   For higher dimensions, we can rely on the n-spherical coordinate system in which the coordinates consist of $d - 1$ angular coordinates $\varphi_1, \ldots, \varphi_{d-2} \in [0, \pi]$ and $\varphi_{d-2} \in [0, 2\pi)$ (Blumenson, 1960). Then we have $\sqrt{|G(\varphi)|} = \sin^{d-2}(\varphi_1) \sin^{d-3}(\varphi_2) \ldots \sin(\varphi_{d-2})$.

Using the ambient cartesian coordinates, the *exponential map* is given by

$$\exp_\mu^c(v) = \cos\left( \sqrt{K}\|v\| \right)\mu + \sin\left( \sqrt{K}\|v\| \right) \frac{v}{\sqrt{K}\|v\|}$$

for all $z \in \mathbb{S}_K^d$ and $v \in \mathcal{T}_z \mathbb{S}_K^d$.

## B   Probability measures on Riemannian manifolds

In what follows, we discuss core concepts of probability measures on Riemannian manifolds and show how naive methods lead to ill- and mis-specified models on manifolds.

Probability measures and random vectors can intrinsically be defined on Riemannian manifolds so as to model uncertainty on non-flat spaces (Pennec, 2006). The Riemannian metric $G(z)$ induces an infinitesimal volume element on each tangent space $\mathcal{T}_z \mathcal{M}$, and thus a measure on the manifold,

$$d\operatorname{Vol}(z) = \sqrt{|G(z)|} \, d\operatorname{Leb},$$

with Leb the Lebesgue measure. Manifold-valued random variables would naturally be characterized by the *Radon-Nikodym derivative* of a measure $\nu$ w.r.t. the Riemannian measure Vol (assuming absolute continuity)

$$p(z) = \frac{d\nu}{d\operatorname{Vol}}(z).$$

### B.1 Ambient Euclidean probability distributions

Unfortunately, conventional probabilistic models implicitly assume a *flat* geometry. This in turn cause these models to either be misspecified or ill-suited to fit manifold distributions. Below we discuss the reasons why.

Let $P_\mathcal{D}$ be a target probability measure that we aim to approximate, and which is defined on a $d$-dimensional manifold $\mathcal{M} \subseteq \mathbb{R}^D$. Furthermore, we assume it admits a Radon-Nikodym derivative $p_\mathcal{D}$ with respect to the manifold invariant measure Vol, denoting $P_\mathcal{D} \ll$ Vol with $\ll$ denoting absolute continuity. Conventional normalizing flows implicitly assume the parametrized probability measure $P_\theta$ to have support on the ambient space $\mathbb{R}^D$ and to be absolutely continuous with respect to the Lebesgue measure $\mathrm{Leb}_{\mathbb{R}^D}$. We denote its density by $p_\theta$.

Next, assume $D = d$, such as for $\mathcal{M} = \mathbb{B}^d \subseteq \mathbb{R}^d$. With $z$ the $d$-dimensional Cartesian coordinates, we have $\frac{d\,\mathrm{Vol}}{d\,\mathrm{Leb}_{\mathbb{R}^d}}(z) = \sqrt{|G(z)|}$. One could then see the manifold-valued target $P_\mathcal{D}$ as being a probability measure on $\mathbb{R}^d$ with a density w.r.t. the Lebesgue measure given by

$$\frac{dP_\mathcal{D}}{d\,\mathrm{Leb}_{\mathbb{R}^d}}(z) = p_\mathcal{D}(z)\sqrt{|G(z)|} \triangleq \tilde{p}_\mathcal{D}(z).$$

In general $P_\mathcal{D} \ll P_\theta$ which implies that the *forward* Kullback-Leibler divergence, or *negative log-likelihood* up to constants, is defined and given by

$$\mathcal{L}^{\text{Like}}(\theta) = D_{\text{KL}}(P_\mathcal{D} \,\|\, P_\theta) + \mathbb{H}(P_\mathcal{D}) = \mathbb{E}_{P_\mathcal{D}}\left[\log\left(\frac{\tilde{p}_\mathcal{D}(z)}{p_\theta(z)}\right)\right] + \mathbb{H}(P_\mathcal{D}) = -\mathbb{E}_{P_\mathcal{D}}\left[\log\frac{p_\theta(z)}{\sqrt{|G(z)|}}\right].$$

Minimising $\mathcal{L}^{\text{Like}}(\theta)$ amounts to pushing-forward $P_\theta$'s mass so that empirical observations $z_i \sim P_\mathcal{D}$ have a positive likelihood under $P_\theta$. Yet, in general the model ($P_\theta$) has (most of his) mass outside the manifold's support which may cause such a *naive* approach to be ill-suited. More crucially it implies that in general the model's mass is *not* covering the full target's support. In that case, the *reverse* Kullback-Leibler divergence $D_{\text{KL}}(P_\theta \,\|\, P_\mathcal{D}) = \mathcal{L}^{\text{KL}}(\theta)$ is not even defined.

Next, consider the case where $\mathcal{M}$ is a submanifold embedded in $\mathbb{R}^D$ with $D > d$, such as $\mathcal{M} = \mathbb{S}^d$ where $D = d + 1$. In this setting the *naive* model $P_\theta$ is even *misspecified* since it is defined on a different probability space than the target. In the limit $\mathrm{supp}(P_\theta) \to \mathcal{M}$, $P_\theta$ is not defined because we have that $\int_{\mathbb{R}^D} P_\theta \to \infty$. The target does consequently not belong to the model's class.

## C   Instantaneous change of variable

In the following we derive the *instantaneous change of density* that a manifold-valued random variable induces when its dynamics are governed by an ODE. We show that in the Riemannian setting this *instantaneous change of density* can be expressed in terms of the manifold's metric tensor.

**Proof of Proposition 2**

*Proof.* For a time dependant particles $z(t)$, whose dynamics are given by the following ODE

$$\frac{dz(t)}{dt} = f(z(t), t)$$

the change in density is given by the Liouville equation (or Fokker–Planck equation without the diffusion term); $\forall z \in \mathcal{M}, \forall t \in [0, T]$

$$\frac{\partial}{\partial t} p(z, t) = -\operatorname{div}(p(z, t)f(z, t))$$

$$= -\left\langle \frac{\partial}{\partial z} p(z, t), f(z, t) \right\rangle_z - p(z, t)\operatorname{div}(f(z, t))$$

where the last step was obtained by applying the divergence product rule. By introducing the time dependence in $z(t)$ and differentiating with respect to time we get

$$\frac{\partial}{\partial t} p(z(t), t) = \left\langle \frac{\partial}{\partial z} p(z(t), t), \frac{\partial}{\partial t} z(t) \right\rangle_{z(t)} + \frac{\partial}{\partial t} p(z(t), t)$$

$$= \left\langle \frac{\partial}{\partial z} p(z(t), t), f(z(t), t) \right\rangle_{z(t)} - \left\langle \frac{\partial}{\partial z} p(z(t), t), f(z(t), t) \right\rangle_{z(t)} - p(z(t), t)\operatorname{div}(f(z(t), t))$$

$$= -p(z(t), t)\operatorname{div}(f(z(t), t))$$

Hence the evolution of the log density is given by

$$\frac{\partial}{\partial t} \log p(z(t), t) = -\operatorname{div}(f(z(t), t)). \tag{8}$$

$\square$

**Divergence computation** For a Riemannian manifold $(\mathcal{M}, g)$, with local coordinates $z$, the divergence of a vector field $f$ is given by

$$\text{div}(f(z,t)) = \frac{1}{\sqrt{|G(z)|}} \sum_{i=1}^{d} \frac{\partial}{\partial z^i} \left( \sqrt{|G(z)|} \, f^i(z,t) \right) \tag{9}$$

$$= \frac{1}{\sqrt{|G(z)|}} \sum_{i=1}^{d} \left( \sqrt{|G(z)|} \, \frac{\partial}{\partial z^i} f^i(z,t) + f^i(z,t) \, \frac{\partial}{\partial z^i} \sqrt{|G(z)|} \right)$$

$$= \sum_{i=1}^{d} \frac{\partial}{\partial z^i} f^i(z,t) + \frac{1}{\sqrt{|G(z)|}} \sum_{i=1}^{d} f^i(z,t) \, \frac{\partial}{\partial z^i} \sqrt{|G(z)|}$$

$$= \text{tr}\left( \frac{\partial}{\partial z} f(z,t) \right) + \frac{1}{\sqrt{|G(z)|}} \left\langle f(z,t), \frac{\partial}{\partial z} \sqrt{|G(z)|} \right\rangle. \tag{10}$$

We note that in Equation 9, $f_i$ are the components of the vector field $f$ with respect to the local unnormalized covariant basis $(\mathrm{e}_i)_{i=1}^{d} = \left( \left( \frac{\partial}{\partial z^i} \right)_z \right)_{i=1}^{d}$. However it is convenient to work with local basis having unit length vectors. If we write $\hat{\mathrm{e}}_i$ for this normalized basis, and $\hat{f}^i$ for the components of $f$ with respect to this normalized basis, we have that

$$f = \sum_i f^i \, \mathrm{e}_i = \sum_i f^i \, \|\mathrm{e}_i\| \frac{\mathrm{e}_i}{\|\mathrm{e}_i\|} = \sum_i f^i \sqrt{G_{ii}} \frac{\mathrm{e}_i}{\|\mathrm{e}_i\|} = \sum_i \hat{f}^i \, \hat{\mathrm{e}}_i$$

using one of the properties of the metric tensor. By dotting both sides of the last equality with the contravariant element $\hat{\mathrm{e}}_i$ we get that $\hat{f}^i = f^i \sqrt{G_{ii}}$. Substituting in Equation 9 yields

$$\text{div}\left( \hat{f}(z,t) \right) = \frac{1}{\sqrt{|G(z)|}} \sum_{i=1}^{d} \frac{\partial}{\partial z^i} \left( \sqrt{\frac{|G(z)|}{G_{ii}(z)}} \, \hat{f}^i(z,t) \right). \tag{11}$$

Combining Equations 8 and 11 and we finally get

$$\frac{\partial \log p(z(t), t)}{\partial t} = -\frac{1}{\sqrt{|G(z)|}} \sum_{i=1}^{d} \frac{\partial}{\partial z^i} \left( \sqrt{\frac{|G(z)|}{G_{ii}(z)}} \, \hat{f}^i(z,t) \right). \tag{12}$$

We rely on this Equation 12 for practical numerical experiments.

# D    regularization

## D.1    $l^2$-norm

Henceforth we motivate the use of an $l^2$ norm regularization in the context of continuous normalizing flows. We do so by highlighting a connection with the dynamical formulation of optimal transport, and by proving that this formulation still holds in the manifold setting.

**Monge-Kantorovich mass transfer problem** Let $(\mathcal{M}, d_{\mathcal{M}})$ be a metric space, and $c : \mathcal{M} \times \mathcal{M} \to [0, +\infty)$ a measurable map. Given probability measures $p_0$ and $p_T$ on $\mathcal{M}$, Monge's formulation of the optimal transportation problem is to find a transport map $\phi^* : \mathcal{M} \to \mathcal{M}$ that realizes the infimum

$$\inf_{\phi} \int_{\mathcal{M}} c(\phi(z), z) \, p_0(dz) \ \text{s.t.} \ p_T \sharp = p_0.$$

It can be shown that this yields a metric on probability measures, and for $c = d_{\mathcal{M}}^2$, it is called the $L^2$ Kantorovich (or Wasserstein) distance

$$d_{W^2}(p_0, p_T)^2 = \inf_{\phi} \int_{\mathcal{M}} d_{\mathcal{M}}(\phi(z), z)^2 \, p_0(dz). \tag{13}$$

By reintroducing the time variable in the $L^2$ Monge-Kantorovich mass transfer problem, the optimal transport map $\phi^*$ can be reformulated as the generated flow from an optimal vector field $f$.

**Proposition 3** (Dynamical formulation from (Benamou and Brenier, 2000))**.** *Indeed we have*

$$d_{W^2}(p_0, p_T)^2 = \inf \frac{1}{T} \int_0^T \|\boldsymbol{f}\|_{p_t}^2 dt = \inf \frac{1}{T} \int_0^T \int_{\mathcal{M}} \langle \boldsymbol{f}(z,t), \boldsymbol{f}(z,t) \rangle_z \, p_t(dz) \, dt \qquad (14)$$

*where the infimum is taken among all weakly continuous distributional solutions of the continuity equation $\frac{\partial}{\partial t} p_t = -\operatorname{div}(p_t \boldsymbol{f})$ such that $p(0) = p_0$ and $p(T) = p_T$. Writing $\phi_t^* = \phi^*(\cdot, t)$ the flow generated by the optimal ODE, then the optimal transport map is given by $\phi^* = \phi_T^*$.*

The RHS of Equation (14) can then be approximated with no extra-cost with a Monte Carlo estimator given samples from $p_t = \phi_t \sharp p_0$.

**Manifold Setting**    Let's now focus on the setting where $\mathcal{M}$ is a Riemannian manifold.

**Proposition 4** (Optimal map (Ambrosio, 2003))**.** *Assume that $\mathcal{M}$ is a $C^3$, complete Riemannian manifold with no boundary and $d_{\mathcal{M}}$ is the Riemannian distance. If $p_0$, $p_T$ have finite second order moments and $p_0$ is absolutely continuous with respect to $\operatorname{vol}_{\mathcal{M}}$, then there exists a unique optimal transport map $\phi$ for the Monge-Kantorovich problem with cost $c = d_{\mathcal{M}}^2$. Moreover there exists a potential $h : \mathcal{M} \mapsto \mathbb{R}$ such that*

$$\phi^*(z) = \exp_z(-\nabla h(z)) \quad \operatorname{vol}_{\mathcal{M}} - a.e..$$

Proposition 3 has been stated and proved for the case $\mathcal{M} = \mathbb{R}^d$. Below we extend the proof given by Benamou and Brenier (2000) for the manifold setting.

*Proof of Proposition 3.*    We follow the same reasoning as the one developed for the Euclidean setting. Let's first upper bound the Wasserstein distance, and then state the optimal flow which yields equality. We have

$$\frac{1}{T} \int_0^T \int_{\mathcal{M}} \|\boldsymbol{f}(z,t)\|_z^2 \, p_t(dz) \, dt = \frac{1}{T} \int_0^T \int_{\mathcal{M}} \|\boldsymbol{f}(\phi(z,t)), t)\|_z^2 \, p_0(dz) \, dt$$

$$= \frac{1}{T} \int_0^T \int_{\mathcal{M}} \left\| \frac{\partial}{\partial t} \phi(z,t) \right\|_z^2 \, p_0(dz) \, dt$$

$$\geq \int_{\mathcal{M}} d_{\mathcal{M}}(\phi(z,T), \phi(z,0))^2 \, p_0(dz) \, dt$$

$$= \int_{\mathcal{M}} d_{\mathcal{M}}(\phi(z,T), z)^2 \, p_0(dz) \, dt$$

$$\geq \int_{\mathcal{M}} d_{\mathcal{M}}(\phi(z), z)^2 \, p_0(dz) \, dt$$

$$= d_{W^2}(p_0, p_T)^2.$$

Thus, the optimal choice of flow $\phi$ is given by

$$\phi(z,t) = \exp_z\left( \frac{t}{T} \log_z(\phi^*(z)) \right), \qquad (15)$$

since $\phi(z, 0) = z$, $\phi(z, T) = \phi^*(z)$ and

$$\left\| \frac{\partial}{\partial t} \phi(z,t) \right\|_z = \left\| \frac{\partial}{\partial t} \phi(z, t = 0) \right\|_z = \left\| \log_z(\phi^*(z) \right\|_z = d_{\mathcal{M}}(\phi^*(z), z).$$

$\square$

Note that the optimal flow from Equation 15 yields integral paths $\gamma(t) = \phi(z,t)$ that are *geodesics* and have constant *velocity*.

**Motivation**    Regularizing the vector field with the RHS of Equation 14 would hence tend to make the generated flow $\phi_T$ closer to the optimal map $\psi^*$. By doing so, one hopes to increase smoothness of $\boldsymbol{f}$ and consequently lower the solver NFE given a fixed tolerance.

This has been observed in the Euclidean setting by Finlay et al. (2020). They empirically showed that regularizing the loss of a CNF with the vector field's $l^2$ norm improves training speed. Motivated by the successful use of gradient regularization (Novak et al., 2018; Drucker and Cun, 1992), they showed that additionally regularizing the Frobenius norm of the vector field's Jacobian helps. In the following subsection we remind that this regularization term can also be motivated from an estimator's variance perspective.

## D.2 Frobenius norm

**Hutchinson's estimator**   Hutchinson's estimator (Hutchinson, 1990) is a simple way to obtain a stochastic estimate of the trace of a matrix. Given a d-dimensional random vector $\epsilon \sim p$ such that $\mathbb{E}[\epsilon] = 0$ and $\mathrm{Cov}(\epsilon) = I_d$, we have

$$\mathrm{tr}(A) = \mathbb{E}_{\epsilon \sim p}[\epsilon^T A \epsilon].$$

Rademacher and Gaussian distributions have been used in practice. For a Rademacher, the variance is given by (Avron and Toledo, 2011)

$$\mathbb{V}_{\epsilon \sim p}[\epsilon^T A \epsilon] = 2 \|A\|_F - 2 \sum_i A_{ii}^2,$$

whereas for a Gaussian it is given by

$$\mathbb{V}_{\epsilon \sim p}[\epsilon^T A \epsilon] = 2 \|A\|_F.$$

**Divergence computation**   As reminded in Appendix C by Equation 10, computing the vector field divergence $\mathrm{div}(f(z, t))$ involves the computation of the trace of vector field's Jacobian $\mathrm{tr}\left(\frac{\partial}{\partial z} f(z, t)\right)$. As highlighted in Grathwohl et al. (2019); Salman et al. (2018), one can rely on the Hutchinson's estimator to estimate this trace with $A = \frac{\partial}{\partial z} f(z, t)$.

The variance of this estimator thus depends on the Frobenius norm of the vector's field Jacobian $\|\frac{\partial}{\partial z} f(z, t)\|_F$, as noted in Grathwohl et al. (2019). Regularizing this Jacobian should then improve training by reducing the variance of the divergence estimator.

# E   Vector flows and neural architecture

Hereafter we discuss about flows generated by vector fields, and neural architectural choices that we make for their parametrization. Properties of vector fields have direct consequences on the properties of the generated flow and in turn on the associated pushforward probability distributions. In particular we derive sufficient conditions on the flow so that it is *global*, i.e. is a bijection mapping the manifold to itself.

## E.1   Existence and uniqueness of a global flow

We start by discussing about vector flows and sufficient conditions on their uniqueness and existence.

**Local flow**   First we remind the Fundamental theorem of flows (Lee, 2003) which gives the existence and uniqueness of a smooth *local flow*.

**Proposition 5** (Fundamental theorem of flows). *Let $\mathcal{M}$ be a smooth complete manifold with local coordinates $z$. Let $f_\theta : \mathcal{M} \times \mathbb{R} \mapsto \mathcal{T}\mathcal{M}$ a $C^1$ time- dependent vector field and $z_0 \in \mathcal{M}$. Then there exists an open interval $I$ with $0 \in I$, an open subset $U \subseteq \mathcal{M}$ containing $z_0$, and a unique smooth map $\phi : I \times U \mapsto \mathcal{M}$ called* local flow *which satisfies the following properties. We write $\phi_t(z) = \phi(z, t)$.*

1. *$\frac{\partial}{\partial t} \phi(z, t) = f_\theta(\phi(z, t), t)$ for all $z, t \in U \times I$, and $\phi_0 = id_\mathcal{M}$.*

2. *For each $t \in I$, the map $\phi_t : U \mapsto \mathcal{M}$ is a local $C^1$-diffeomorphism.*

Note that with such assumptions, the existence and uniqueness of flows $\phi_t$ are only *local*.

**Global flow**   We would like the flow $\phi$ to be defined for all times and on the whole manifold, i.e. a *global flow* $\phi : \mathcal{M} \times \mathbb{R} \mapsto \mathcal{M}$. Fortunately, if $\mathcal{M}$ is *compact* (such as n-spheres and torii), then the flow is global (Lee, 2003). We show below that another sufficient condition for the flow to be global is that the vector field be bounded.

**Proposition 6** (Global Flow). *Let $\mathcal{M}$ be a smooth complete manifold. Let $f_\theta : \mathcal{M} \times \mathbb{R} \mapsto \mathcal{T}\mathcal{M}$ be a $C^1$ bounded time-dependent vector field. Then the domain of the flow $\phi$ is $\mathbb{R} \times \mathcal{M}$, i.e. the flow is global.*

**Corollary 6.1.** *For each $t \in \mathbb{R}$, the map $\phi_t : \mathcal{M} \mapsto \mathcal{M}$ is a $C^1$-diffeomorphism.*

*Proof of Proposition 6.* Let $c > 0$ s.t. $\|f\| < c$, and $z_0 \in \mathcal{M}$ be an initial point. Proposition 5 gives the existence of an open interval $I = (a, b)$, a neighbourhood $U$ of $z_0$ and a local flow $\phi : (a, b) \times U \mapsto \mathcal{M}$. We write $\gamma = \phi(z_0, \cdot)$. The maximal interval of $\gamma$ is $(a, b)$, which means that $\gamma$ cannot be extended outside $(a, b)$. Suppose that $b < \infty$.

The integral path $\gamma$ is Lipschitz continuous on $(a, b)$ since we have

$$d_{\mathcal{M}}(\gamma(t), \gamma(s)) \le \int_s^t \|\gamma'(t)\| \, dt = \int_s^t \|f(\gamma(t), t)\| \, dt \le c \, |t - s| \tag{16}$$

for all $s < t \in (a, b)$.

Let $(t_n)$ be a sequence in $(a, b)$ that converges to $b$. Then since $(t_n)$ is a convergent sequence, it must also be a Cauchy sequence. Then $\gamma(t_n)$ is also a Cauchy sequence by Equation 16. Since $\mathcal{M}$ is geodesically complete, it follows by Hopf-Rinow theorem that $(\mathcal{M}, d_{\mathcal{M}})$ is complete, hence that $\gamma(t_n)$ converges to a point $p \in \mathcal{M}$.

Now suppose that $(s_n)$ is another sequence in $(a, b)$ that converges to $b$. Then by Equation 16 $\lim_{n \to \infty} d(\gamma(s_n), \gamma(t_n)) = 0$, thus $\gamma(s_n)$ also converges to $\lim_{n \to \infty} \gamma(t_n) = p$. So for every sequence $(t_n)$ in $(a, b)$ that converges to $b$, we have that $(\gamma(t_n))$ converges to $p$. Therefore by the sequential criterion for limits, we have that $\gamma$ has the limit $p$ at the point $b$. Therefore, define $\gamma(b) = p$ and so $\gamma$ is continuous at $b$ which is a contradiction. $\square$

## E.2 Geodesic distance layer

The expressiveness of CNFs directly depends on the expressiveness of the vector field and consequently on its architecture. Below we detail and motivate the use of a *geodesic distance* layer, as an input layer for the vector field neural architecture.

**Linear layer** A linear layer with one *neuron* can be written in the form $h_{a,p}(z) = \langle a, z - p \rangle$, with orientation and offset parameters $a, p \in \mathbb{R}^d$. Stacking $l$ such neurons $h$ yields a linear layer with width $l$. This neuron can be rewritten in the form

$$h_{a,p}(z) = \text{sign}\left(\langle a, z - p \rangle\right) \|a\| \, d_E\left(z, H_{a,p}^K\right)$$

where $H_{a,p} = \{z \in \mathbb{R}^p \mid \langle a, z - p \rangle = 0\} = p + \{a\}^\perp$ is the decision hyperplane. The third term is the distance between $z$ and the decision hyperplane $H_{a,p}^K$ and the first term refers to the side of $H_{a,p}^K$ where $z$ lies.

**Poincaré ball** Ganea et al. (2018) analogously introduced a neuron $f_{a,p}^K : \mathbb{B}_K^d \to \mathbb{R}^p$ on the Poincaré ball,

$$h_{a,p}^K(z) = \text{sign}\left(\left\langle a, \log_p^K(z) \right\rangle_p\right) \|a\|_p \, d^K(z, H_{a,p}^K) \tag{17}$$

with $H_{a,p}^K = \left\{z \in \mathbb{B}_K^d \mid \left\langle a, \log_p^K(z) \right\rangle = 0\right\} = \exp_p^K(\{a\}^\perp)$. A closed-formed expression for the distance $d^K(z, H_{a,p}^K)$ was also derived, $d^K(z, H_{a,p}^K) = \frac{1}{\sqrt{|K|}} \sinh^{-1}\left(\frac{2\sqrt{|K|}|\langle -p \oplus_K z, a \rangle|}{(1 + K\|-p \oplus_K z\|^2)\|a\|}\right)$ in the Poincaré ball. To avoid an over-parametrization of the hyperplane, we set $p = \exp_0(a_0)$, and $a = \Gamma_{0 \to p}(a_0)$ with $\Gamma$ parallel transport (under Levi-Civita connection). We observed that the term $\|a\|_p$ from Equation 17 was sometimes causing numerical instabilities, and that when it was not it also did not improve performance. We consequently removed this scaling term. The hyperplane decision boundary $H_{a,p}^K$ is called *gyroplane* and is a semi-hypersphere orthogonal to the Poincaré ball's boundary.

**Hypersphere** In hyperspherical geometry, geodesics are great circles which can be parametrized by a vector $w \in \mathbb{R}^{d+1}$ as $H_w = \{z \in \mathbb{S}^d \mid \langle w, z \rangle = 0\}$. The geodesic distance between $z \in \mathbb{S}^d$ and the hyperplane $H_w$ is then given by

$$d(z, H_w) = \left| \sin^{-1}\left(\frac{\langle w, z \rangle}{\sqrt{\langle w, w \rangle}}\right) \right|.$$

In a similar fashion, a neuron is now defined by

$$h_w(z) = \|w\|_2 \sin^{-1}\left(\frac{\langle w, z \rangle}{\sqrt{\langle w, w \rangle}}\right).$$

**Geodesic distance layer** One can then horizontally-stack $l$ neurons to make a *geodesic distance* layer $g : \mathcal{M} \mapsto \mathbb{R}^l$ (Mathieu et al., 2019). Any standard feed-forward neural network can then be vertically-stacked on top of this layer.

# F    Extensions

## F.1    Product of manifolds

Having described CNFs for complete smooth manifolds in Section 2, we extend these for product manifolds $\mathcal{M} = \mathcal{M}_1 \times \cdots \times \mathcal{M}_k$. For instance a $d$-dimensional torus is defined as $\mathbb{T}^d = \underbrace{\mathbb{S}^1 \times \cdots \times \mathbb{S}^1}_{d}$.

Any density $p_\theta(z_1, \ldots, z_K)$ can decomposed via the chain rule of probability as

$$p_\theta(z_1, \ldots, z_K) = \prod_k p_{\theta_k}(z_k \mid z_1, \ldots, z_{k-1})$$

where each conditional $p_{\theta_k}(z_k \mid z_1, \ldots, z_{k-1})$ is a density on $\mathcal{M}_k$. As suggested in Rezende et al. (2020), each conditional density can be implemented via a flow $\phi_k : \mathcal{M}_k \mapsto \mathcal{M}_k$ generated by a vector field $f_k$, whose parameters $\theta_k$ are a function of $(z_1, \ldots, z_{k-1})$. Such a flow $\phi = \phi_1 \circ \cdots \circ \phi_k$ is called *autoregressive* (Papamakarios et al., 2017) and conveniently has a lower triangular Jacobian, which determinant can be computed efficiently as the product of the diagonal term.

# G    Experimental details

Below we fully describe the experimental settings used to generate results introduced in Section 4.

**Architecture** The architecture of the *vector field* $f_\theta$ is given by a multilayer perceptron (MLP) with 3 hidden layers and 64 hidden units – as in (Grathwohl et al., 2019) – for *projected* (e.g. stereographic and wrapped cf Section 3) and *naive* (cf Appendix B.1) models. We rely on tanh activation. For our Riemmanian continuous normalizing flow (RCNF), the input layer of the MLP is replaced by a *geodesic distance* layer (Ganea et al., 2018; Mathieu et al., 2019) (see Appendix E.2).

**Objectives** We consider two objectives, a Monte Carlo (MC) estimator of the negative log-likelihood

$$\hat{\mathcal{L}}^{\text{Like}}(\theta) = - \sum_{i=1}^{B} \log p_\theta(z_i) \text{ with } z_i \sim P_\mathcal{D}$$

and a MC estimator of the reverse KL divergence

$$\hat{\mathcal{L}}^{\text{KL}}(\theta) = \sum_{i=1}^{B} \log p_\theta(h_\theta(\epsilon_i)) - \log p_\mathcal{D}(h_\theta(\epsilon_i))$$

with $z_i \sim P_\theta$ being reparametrized as $z_i = h_\theta(\epsilon_i)$ and $\epsilon_i \sim P$.

**Optimization** All models are trained by the stochastic optimizer Adam (Kingma and Ba, 2015) with parameters $\beta_1 = 0.9, \beta_2 = 0.999$, batch-size of 400 data-points and a learning rate set to $1e^{-3}$.

**Training** We rely on the Dormand-Prince solver (Dormand and Prince, 1980), an adaptive Runge-Kutta 4(5) solver, with absolute and relative tolerance of $1e-5$ to compute approximate numerical solutions of the ODE. Each solver step is projected onto the manifold. Models are trained on a cluster of GeForce RTX 2080 Ti GPU cards.

## G.1    Hyperbolic geometry and limits of conventional and wrapped methods

In this experiment the target is set to be a wrapped normal on $\mathbb{B}^2$ (Nagano et al., 2019; Mathieu et al., 2019) with density $\mathcal{N}^{\text{W}}(\exp_0(\alpha \, \partial x), \Sigma) = \exp_{\mu\sharp} \mathcal{N}(\alpha \, \partial x, \Sigma)$ with $\Sigma = \text{diag}(0.3, 1.0)$. The scalar parameter $\alpha$ allows us to locate the target closer or further away from the origin of the disk. Through this experiment we consider three CNFs:

- *Naive*: $P_\theta^{\mathrm{N}} = \phi_\sharp^{\mathbb{R}^2} \mathcal{N}(0,1)$

- *Wrapped*: $P_\theta^{\mathrm{W}} = (\exp_{\mathbf{0}} \circ \phi^{\mathbb{R}^2})_\sharp \mathcal{N}(0,1)$

- *Riemannian*: $P_\theta^{\mathrm{R}} = \phi_\sharp^{\mathbb{B}^2} \mathcal{N}^{\mathrm{W}}(0,1)$

with $\phi^{\mathbb{R}^2}$ a conventional CNF on $\mathbb{R}^2$, $\phi^{\mathbb{B}^2}$ our RCNF introduced in Section 2, $\mathcal{N}(0,1)$ the standard Gaussian and $\mathcal{N}^{\mathrm{W}}(0,1)$ the standard wrapped normal. For the RCNF we scale the vector field as

$$\boldsymbol{f}_\theta(\boldsymbol{z}) = |G(\boldsymbol{z})|^{-1/2} \, \texttt{neural\_net}(\boldsymbol{z}) = \left(\frac{1 - \|\boldsymbol{z}\|^2}{2}\right)^2 \texttt{neural\_net}(\boldsymbol{z}).$$

These three models are trained for 1500 iterations, by minimizing the negative log-likelihood (see Figure 4). The reported results are averaged over 12 runs. When training, the divergence is approximated by the (Hutchinson) stochastic estimator from Equation 5.

### G.2 Spherical geometry

Through the following spherical experiments we consider the two following models

- *Stereographic*: $P_\theta^{\mathrm{S}} = (\rho^{-1} \circ \phi^{\mathbb{R}^2})_\sharp \mathcal{N}(0,1)$

- *Riemannian*: $P_\theta^{\mathrm{R}} = \phi_\sharp^{\mathbb{S}^2} \mathcal{U}(\mathbb{S}^2)$

with $\rho^{-1}$ the inverse of the stereographic projection, $\phi^{\mathbb{R}^2}$ a conventional CNF on $\mathbb{R}^2$, $\phi^{\mathbb{S}^2}$ our RCNF, $\mathcal{N}(0,1)$ the standard Gaussian and $\mathcal{U}(\mathbb{S}^2)$ the uniform distribution on $\mathbb{S}^2$. For the RCNF we project the output layer of the vector field as

$$\boldsymbol{f}_\theta(\boldsymbol{z}) = \mathrm{proj}_{\mathcal{T}_{\boldsymbol{z}}\mathbb{S}^2} \, \texttt{neural\_net}(\boldsymbol{z}) = \frac{\texttt{neural\_net}(\boldsymbol{z})}{\|\texttt{neural\_net}(\boldsymbol{z})\|^2}$$

so as to enforce output vectors to be *tangent*. All spherical experiments were performed using the exact divergence estimator.

**Limits of the stereographic projection model** In this experiment the target is chosen to be a vMF$(\boldsymbol{\mu}, \kappa)$ located at $\boldsymbol{\mu} = -\boldsymbol{\mu}_0$ with $-\{\mu_0\} = (-1, 0, \ldots, 0)$. Both models are trained for 3000 iterations by minimizing the negative log-likelihood and the reverse KL divergence. The reported results are averaged over 4 runs.

**Density estimation of spherical data** Finally we consider four earth location datasets, representing respectively volcano eruptions (NOAA, 2020b), earthquakes (NOAA, 2020a), floods (Brakenridge, 2017) and wild fires (EOSDIS, 2020). The reported results are averaged over 12 runs.

Concerning the CNFs, these models are trained by minimizing the negative log-likelihood for 1000 epochs, except for the volcano eruption dataset where 3000 epochs are required for convergence. We observed that training models with a solver's tolerance of $1e-5$ was computationally intensive so we lowered the training tolerance to $1e-3$ for the volcano eruptions, earthquakes and wild fires datasets, and $1e-4$ for the floods dataset, while keeping it to $1e-5$ during evaluation. We additionally did not use the geodesic layer (presented in Appendix E.2) to lower the computational time. We also observed that annealing the learning rate such that $\alpha(t) = 0.98^{(t/300)}\alpha_0$ with $\alpha_0 = 1e-3$ helped stabilizing training convergence.

Concerning the mixture of von Mises-Fisher distributions, the parameters are learned by minimizing the negative log-likelihood with Riemannian Adam (Becigneul and Ganea, 2019). The number of epochs is set to 10000 for all datasets but for the volcano eruption one where 30000 epochs are required for the vMF model to converge. The learning rate and number of mixture components are selected by performing a hyperparameter grid search, over the following range: learning rate $\in \{1e-1, 5e-2, 1e-2\}$ and number of components $\in \{50, 100, 150, 200\}$.

## H  Additional figures

Figure 7: Ablation study of the geodesic layer computational impact for the *Riemannian* model. Negative Log-likelihood of *Riemannian* CNFs trained to fit a $\mathcal{N}^{\mathrm{W}}(\exp_{\mathbf{0}}(\alpha\,\partial x), \Sigma)$ target on $\mathbb{B}^2$.

Figure 8: Density estimation for earth sciences data with Robinson projection. Blue and red dots represent training and testing datapoints, respectively. Heatmaps depict the log-likelihood of the trained models.