[Reviews · NeurIPS 2020]

Review 1

Summary and Contributions: This paper proposes Riemannian continuous normalizing flows (CNFs), which is more adequate for non-Euclidean data, such as directional data lying on hyperspheres, than conventional CNFs. The key idea of the proposed method is to take into account the underlying data geometry both when specifying and solving the associated ordinary differential equation, to ensure the faithfulness of the flow to the data manifold. For simplicity, the authors focus on constant curvature manifolds and evaluate their method on data living on spheres and hyperbolic spaces.

Strengths: Developing probabilistic methods for modeling data living in non-Euclidean spaces is an important research topic, as there are many real-world applications where such data can be encountered. This paper makes an interesting contribution in this direction in the context of continuous normalizing flows. The proposed method is principled and theoretically sound. Numerical experiments, even if they a bit proof-of-concept, illustrate the benefit of the proposed Riemannian CNFs for modeling data lying on spheres and hyperbolic space.

Weaknesses: The limitations (e.g., scalability) of the proposed methods are not discussed. The experiments on synthetic datasets involve relatively simple (unimodal) target distributions. It would be interesting to consider more complicated distributions as this is the setting in which it makes more sense to rely on normalizing flows. Comparisons can be improved by considering stronger baselines (e.g., [1]). [1] Rezende, Danilo Jimenez, et al. "Normalizing flows on tori and spheres." arXiv preprint arXiv:2002.02428 (2020).

Correctness: Yes to the extent that I checked.

Clarity: Yes.

Relation to Prior Work: Yes.

Reproducibility: Yes

Additional Feedback: * Update after rebuttal After reading the authors’ response and other reviews, my original impressions remain the same. I encourage the authors to take into account the reviewers feedback and improve the experiments as promised in their rebuttal.


Review 2

Summary and Contributions: The authors extend continuous normalizing flows to the manifold-valued setting. They achieve this by 1. deriving a change of variable formula that holds in the manifold setting (as the commonly used one does not), 2. slightly modifying the vector field neural network's architecture to ensure that manifold constraints are satisfied (they do this using pre-existing methods), and 3. using a different ODE solver which ensures that the manifold constraints are once again satisfied (which is also a pre-existing ODE solver).

Strengths: I enjoyed reading the paper, and believe it has several points in its favor. From a theoretical perspective, I find the derivation of the change of variable formula in manifolds to be a valuable contribution to the machine learning community, and think it could encourage future work towards learning the manifold. Additionally, while limited to low-dimensional data, I found the experiments to support the conclusion that this method works better than other normalizing flow alternatives and think it could become the standard method for sphere or torus-valued data.

Weaknesses: In my view, the main disadvantage of the paper is that its scope is somewhat narrow: when used to directly model observations, the author's method can only be applied when the manifold is known beforehand. High dimensional data of interest in machine learning will often lie in a low dimensional manifold (e.g. natural images), but the manifold is typically not known in advance. That being said, the problem of learning the manifold is a much harder one, and this complaint is a bit of a nitpick on my part, as I think the change of variable formula that the authors derive could be a stepping stone in this direction. Another complaint is that, while the authors convincingly show that their method performs better than other normalizing flows on spherical data, the data is essentially 2-dimensional, so it is not clear to me that a neural network model will necessarily be the best performing alternative. For example, did you consider comparing against a mixture of vMF distributions? Finally, I think the absence of experiments in a VAE setting are a shortcoming of the paper. There is a recent line of work endowing the latent space of VAEs with a manifold structure, e.g. [1, 2, 3]. While these papers are cited, comparing against them in a VAE setting would significantly strengthen the experiments section of the paper, as it would show empirically that the proposed method is not only useful to model manifold-valued data. Could you include such additional experiments? [1] Nagano et al., A Wrapped Normal Distribution on Hyperbolic Space for Gradient-Based Learning, ICML 2019 [2] Mathieu et al., Continuous Hierarchical Representations with Poincaré Variational Auto-Encoders, NeurIPS 2019 [3] Bose et al., Latent Variable Modelling with Hyperbolic Normalizing Flows, ICML 2020

Correctness: I did not find any errors in the claims or empirical methodology followed by the authors.

Clarity: The paper is excellently written. My only suggestion is to better reference the discussion in appendix B within the main manuscript. One should not compare manifold-valued densities against R^D valued ones as they are not Radon-Nikodym derivatives with respect to the same base measure. I believe non-careful readers might be tempted to make such comparisons, and these comparisons should be more clearly warned against in the main manuscript.

Relation to Prior Work: Prior work is adequately discussed.

Reproducibility: Yes

Additional Feedback: How much of a computational constraint is having to project the outputs of the network to the tangent space, along with having to use a projected ODE solver? Could you provide a wall-clock time comparison between your method and the competing alternatives? =============================================================== UPDATE AFTER REBUTTAL =============================================================== After reading the author's response, my assessment of the paper remains the same.


Review 3

Summary and Contributions: The paper introduces a Continuous Normalizing Flow which is aware of the (Riemannian) manifold structure it is defined on. Their key contribution is to express the instantaneous change of variables using a Riemannian metric. They leverage previous developments to efficiently compute it and evaluate the flow. Since several phenomena are naturally represented by data on Riemannian manifolds such as spheres, torii, and hyperbolic spaces, this is a valuable contribution to density estimation on those domains.

Strengths: The paper has a solid theoretical development, providing detailed derivations for key steps, with adequate justification. I also appreciate how general the formulation is. In contrast to previous approaches, it does not rely on specific transformations designed for the manifold at hand. I see the approach as an important and relatively novel contribution to density estimation on Riemannian manifolds.

Weaknesses: In my opinion, the experiments are the weakest aspect of the paper. First, the experiment showing limitations of the stereographic projection model seem unnecessary to me, as the stereographic projection is known to lead to problems in its singular point. Such singular points can even be changed if needed by the application, resulting in slightly different transformations. Even so, a lot of space is dedicated to show this and that the proposed method is not affected by it. It strikes me as a missed opportunity for more experiments on the applicability of the proposed method on real data. Furthermore, experiments concerning real data seemed not very informative of the model's applicability, since: (i) all data sets are of similar nature, (ii) small, and (iii) their distributions do not seem challenging to model, as indicated by the performance attained by the baseline. I would be eager to increase my score for the paper if an additional experiment is performed on a higher-dimensional problem. As an added benefit, this could also be an opportunity to showcase advantages with regards to the mentioned numerical instability of prior approaches. Possible other domains that could be explored are even mentioned in the paper itself: * "The shape of proteins can be parametrized using torii (Hamelryck et al., 2006)" * "Cell developmental processes can be described through paths in hyperbolic space (Klimovskaia et al., 2019)" * "Human actions can be recognized in video using matrix manifolds (Lui, 2012)" One can also artificially devise a complex high-dimensional data set on a manifold by projecting high-dimensional data from Euclidean space onto a Riemannian manifold and then attempting to model the resulting distribution.

Correctness: The claims and method are correct. Empirical methodology needs some improvement, as mentioned previously.

Clarity: The paper is very well-written, with an easy to follow argumentation, justification, and development. One problem is that the authors constantly point towards supplementary material for any depth in the discussion. However, I understand the difficulty of addressing this issue with the limited space available.

Relation to Prior Work: Very good discussion of prior work in normalizing flows.

Reproducibility: Yes

Additional Feedback: # After reading the author response: As the authors commit themselves to do an additional experiment on higher-dimensional data, I no longer see the experiments as a major issue to be addressed. Hence, I believe this paper could be accepted in this case.


Review 4

Summary and Contributions: The paper proposes a normalizing flow method for Riemannian manifolds, which ensures that the flow is always located on the manifold for any starting sampled particles. The flow function is defined via vector fields on manifolds and approximated by the associated ODE.

Strengths: The idea in this paper have shown great novelty, the claims in this paper are rigorously proved in the view of manifolds. The experiments part have shown good results, figures for the tests on the earth sciences data are beautiful.

Weaknesses: All the experiments in this paper are tested on regular manifolds (the earth's surface is approximated as a perfect sphere), this part would be strenghtened if more experiments on complicated or high-genus manifolds are provided.

Correctness: The claims in this paper are correct, and the apsects of continuous normalizing flows on Riemannian manifolds are defined clearly and proved in detail.

Clarity: The paper is well-written and readable.One small flaw is some parts in section 1 and 2 are repetitive.

Relation to Prior Work: This work extended the traditional normalizing flow to general manifolds, and have better performance than some of the previous works, for example, stereographic mathods. Besides flow method, there are other method, such as optimal transport mapping method, to transform one distribution to the other. The authors neglected this direction.

Reproducibility: Yes

Additional Feedback: Flow method is commonly used in the literature of optimal transport. One of the major drawbacks of this method is that it can not model the transformation for a white noise to multi-mode distributions, because in this situation, the mapping is not continuously globally. This will cause mode collapse in DL systems. The authors need to address this problem. [POST REBUTTAL] I thank the authors for the detailed response. In fact, even if the target measure support is simply connected, the transport map may not be continuous either. In general, I think the work is promising and remain the same score.

[Author Response · NeurIPS 2020]

We thank the reviewers for their time, helpful feedback, and advice. We are pleased that overall, reviewers praised the
clarity, rigour and contributions of the work. We are encouraged that reviewers acknowledged novelty (R3, R4) and
appreciated our work as a principled contribution in the development of machine learning methods for non-Euclidean
data (R1, R2). We thank reviewers for their kind words, and hope to address any remaining concerns below.

**(R1, R2, R3) Uni-modal distributions.** We thank the reviewers for raising this question. The goal of our two
experiments with unimodal target distributions is to demonstrate specific pathologies of previously introduced methods.
We believe this is most clearly demonstrated by a unimodal distribution at the point (or the limit) of the pathology as it
removes additional modelling artefacts that are introduced through more complex distributions. We further evaluate the
capacity of our method to model highly multi-modal distributions in our real-world experiments in which we can show
substantial improvements to baselines. We will clarify the different purpose of these experiments in the paper.

We agree with (R2) that "the stereographic projection is known to lead to problems in its singular point." Yet, current
SOTA methods are actively using such a parametrization (e.g., Gemici et al. (2016)) and we believe it is important
to show and evaluate this aspect. Moreover, most manifolds of interest similarly have a non-Euclidean topology, so
applying previous *projected* methods on these manifolds would also yield a similar pathology.

**(R2, R3) Real-world data.** Our experiments on real-world data are motivated from problems in climate and earth
science, hence leading to empirical assessments on $\mathbb{S}^2$. We believe these experiments to be informative since they show
that our method is a) scalable and b) can fit highly complex and multi-modal distributions more accurately than previous
methods. Moreover, please note that our model can straightforwardly be applied on higher dimensional manifolds.
We agree with reviewers that our model would be strengthened by an additional experiment on a high dimensional
manifold. To achieve this we will run a fourth experiment by first computing hyperbolic multi-dimensional embeddings
of WordNet graph data, and then fitting our model to the obtained empirical distribution.

**(R4) Optimal transport (OT) and flows.** As reminded by (R4), the field of OT is core for the study of flow – known
as *transport map* – transforming one distribution into another. Indeed, we develop this connection in Appendix D.1
and show that the dynamical formulation of OT still holds in the manifold setting. Regarding (R4)'s concern that our
method "can not model the transformation for a white noise to multi-mode distributions": Our method is theoretically
sound as it able to model multi-modal distributions as long as supports are connected (see Cornish et al. (2019, Theorem
2.1)), and is empirically shown to model well multi-modal earth data (cf Table 3 and Figure 6).

**(R1, R2) Computational aspects.** We thank (R1) for suggesting to expand on the limitations of our method. As
reminded by (R2), projections required by our approach can increase the computational cost. With our current
implementation, we empirically find that this additional cost amounts to $\sim 20\%$ for the Poincaré ball and $\sim 30\%$ for
$\mathbb{S}^2$. This cost can be further reduced by only projecting the output of the ODE solver steps. We thank (R2) for the
suggestion and we will rigorously compute and include wall-clock time comparisons in the next draft.

**(R1, R2) Empirical comparison to related methods.** We thank (R2) for suggesting the mixture of von Mises-Fisher
(vMF) baseline. Indeed, we found in our early empirical assessments that high multi-modality (e.g., as occurring in the
different the earth datasets) would prevent a mixture of vMF distributions from being a competitive baseline. We will
re-run this baseline and include its performance in Table 3.

We also agree with (R1) that comparing our method to Rezende et al. (2020) would indeed be valuable. Unfortunately,
the code and necessary experimental details have not yet been released, preventing us from a detailed comparison.

**(R2, R4) Additional manifolds.** We agree with (R4) that applying the proposed method to more exotic manifolds
is an exciting direction. We are currently exploring applications on several Lie groups such as orthogonal or positive
definite matrices. As reminded by (R2) we indeed assume that the manifold is known beforehand. We agree that
learning the manifold is an exciting task, but indeed a harder one as shown by the field of topological data analysis.

**(R2) VAE experiments.** We thank (R2) for suggesting to leverage our model in a VAE setting. We agree that this is
indeed a promising application of our method (e.g., see also Bose et al. (2020)) and plan to explore this in future work.

**(R2) Naive Euclidean method.** We thank (R2) for highlighting the potential risk for non-careful readers to compare
manifold-valued densities against $\mathbb{R}^D$ valued ones. We updated the introduction to better refer this.

## References

Bose, A. J., Smofsky, A., Liao, R., Panangaden, P., and Hamilton, W. L. (2020). Latent Variable Modelling with
Hyperbolic Normalizing Flows. *arXiv:2002.06336 [cs, stat]*.
Cornish, R., Caterini, A. L., Deligiannidis, G., and Doucet, A. (2019). Relaxing bijectivity constraints with continuously
indexed normalising flows. *arXiv preprint arXiv:1909.13833*.
Gemici, M. C., Rezende, D., and Mohamed, S. (2016). Normalizing Flows on Riemannian Manifolds. *arXiv:1611.02304
[cs, math, stat]*.


[Meta-Review · NeurIPS 2020]

The paper proposes a ODE normalizing flow for Riemannian manifolds. The presentation is very clear. A few issues with the experimental results have been raised.